# Genomic epidemiology reveals geographical clustering of multidrug-resistant *Escherichia coli* ST131 associated with bacteraemia in Wales

Rhys T. White [1,2,3,10], Matthew J. Bull[4,5], Clare R. Barker[4], Julie M. Arnott[6], Mandy Wootton[5], Lim S. Jones[5], Robin A. Howe[5], Mari Morgan [6], Melinda M. Ashcroft [7], Brian M. Forde [2,8], Thomas R. Connor [4,9] ✉ & Scott A. Beatson [1,2,3] ✉

Antibiotic resistance is a significant global public health concern. Uropathogenic *Escherichia coli* sequence type (ST)131, a widely prevalent multidrug-resistant clone, is frequently associated with bacteraemia. This study investigates third-generation cephalosporin resistance in bloodstream infections caused by *E. coli* ST131. From 2013-2014 blood culture surveillance in Wales, 142 *E. coli* ST131 genomes were studied alongside global data. All three major ST131 clades were represented across Wales, with clade C/*H*30 predominant (*n* = 102/142, 71.8%). Consistent with global findings, Welsh strains of clade C/*H*30 contain $\beta$-lactamase genes from the $bla_{\text{CTX-M-1}}$ group (*n* = 65/102, 63.7%), which confer resistance to third-generation cephalosporins. Most Welsh clade C/*H*30 genomes belonged to sub-clade C2/*H*30Rx (58.3%). A Wales-specific sub-lineage, named GB-WLS.C2, diverged around 1996-2000. An introduction to North Wales around 2002 led to a localised cluster by 2009, depicting limited genomic diversity within North Wales. This investigation emphasises the value of genomic epidemiology, allowing the detection of genetically similar strains in local areas, enabling targeted and timely public health interventions.

Uropathogenic *Escherichia coli* (UPEC) are the leading cause of urinary and systemic infections. UPEC present an increasing burden to public health due to increasing antimicrobial resistance (AMR). Increasing rates of AMR can lead to treatment failures and progression to systemic bacteraemia infections. Additionally, the emergence and dissemination of UPEC strains encoding AMR are causing economic damage to countries and healthcare systems[1,2]. Incidences of *E. coli*-associated bacteraemia are increasing globally. In Wales (known as

[1]School of Chemistry and Molecular Biosciences, The University of Queensland, Brisbane, QLD 4072, Australia. [2]Australian Infectious Disease Research Centre, The University of Queensland, Brisbane, QLD 4072, Australia. [3]Australian Centre for Ecogenomics, The University of Queensland, Brisbane, QLD 4072, Australia. [4]Microbiomes, Microbes and Informatics Group, Organisms and Environment Division, School of Biosciences, Cardiff University, Cardiff CF10 3AX, United Kingdom. [5]Public Health Wales Microbiology, University Hospital of Wales, Cardiff, Wales CF14 4XW, United Kingdom. [6]Healthcare Associated Infection, Antimicrobial Resistance & Prescribing Programme (HARP), Public Health Wales, 2 Capital Quarter, Tyndall Street, Cardiff, Wales CF10 4BZ, United Kingdom. [7]Department of Microbiology and Immunology, The University of Melbourne at The Peter Doherty Institute for Infection and Immunity, Melbourne, VIC, Australia. [8]The University of Queensland, UQ Centre for Clinical Research (UQCCR), Royal Brisbane & Women's Hospital Campus, Brisbane, QLD 4029, Australia. [9]Public Health Genomics Programme, Public Health Wales, 2 Capital Quarter, Tyndall Street, Cardiff, Wales CF10 4BZ, United Kingdom. [10]Present address: Health Group, Institute of Environmental Science and Research, 5022 Porirua, New Zealand. ✉e-mail: connortr@cardiff.ac.uk; s.beatson@uq.edu.au

Cymru in the Welsh language), the five-year rolling average age-standardised mortality for deaths involving *E. coli* bacteraemia (ECB) almost doubled from 4.0 (95% confidence interval (CI): 2.3 to 6.4) per one million population in 2002–06, to 7.7 (95% CI: 5.4 to 10.6) in 2006–10 (Supplementary Materials, Figure S1)[3]. Previous studies have shown that most urinary tract infections (UTIs) are caused by a limited number of key UPEC clonal lineages including sequence types (ST)131, ST69, ST73, ST95 and ST12[4–6]. UPEC are predominantly found within the *E. coli* phylogenetic groups B1, B2, or D.

*E. coli* ST131 is a high-risk pandemic clone that is frequently associated with bacteraemia[7] and UTIs, and is a major circulating lineage in the United Kingdom (UK)[8]. The phylogeny of ST131 is characterised by three major clades. Clade A (O16:H5 serotype) is the most divergent and primarily defined by the type 1 fimbriae *fimH* adhesin *H*41 allele, Clade B (O25b:H4 serotype) is frequently associated with animal to human transmission and primarily defined by *fimH* allele *H*22 and Clade C (O25b:H4 serotype) represents the largest ST131 clade and is primarily defined by the *fimH* allele *H*30[9–12]. Clade C can be further delineated into two major sub-lineages: C1 (*H*30R) and C2 (*H*30Rx), which are resistant to fluorquinolones, with C2 strains frequently carrying the CTX-M-15 extended-spectrum $\beta$-lactamase (ESBL) allele[9,11]. Phylogenomic analyses estimated that Clade C ST131 emerged in North America in either the 1980s[10,12] or 1990s[13]. The successful transmission of ST131 globally is attributed to: (i) resistance to many treatments by the carriage of genes encoding resistance to multiple antimicrobial agents;[9,10,14] (ii) the ability to cause disease that other opportunistic or commensal strains do not possess through pathogenicity, fitness, and metabolic factors;[15–17] (iii) the ability to survive in human serum[18] due to capsule production;[19] (iv) transmission in various environments including healthcare- and community-acquired transmission[20], and (v) the open pangenome and low genomic fluidity associated with ST131, which is rare in *E. coli* and the diversification of certain genetic loci working under the negative frequency-dependent selection model[21]. ST131 can colonise and persist in hosts for extended periods causing recurrent UTIs, typically within one year of the initial infection[22]. ST131 cases frequently harbour resistance to broad-spectrum therapies such as third-generation cephalosporins (3GCs)[23,24] and fluoroquinolones[25]. In ST131, the carriage of ESBLs facilitates the principal resistance mechanism to 3GCs.

In 2018, the UK National Institute for Clinical Excellence (NICE) issued guidelines concerning acute pyelonephritis[26]. This recommended urine culture susceptibility testing and promoted the use of several $\beta$-lactams, trimethoprim, ciprofloxacin (fluoroquinolone), or amoxicillin and clavulanic acid as first-line antibiotics. This could be contributing to increasing rates of ESBL-producing *E. coli* across the UK[27]. Between 2017-2018, data from England and Wales showed that at least 14.1% ($n = 4950/35,050$)[28] and 13.3% ($n = 354/2,663$)[29] of *E. coli* bloodstream isolates presented resistance to 3GCs, respectively. In England, this translates to approximately 5,000 annual cases, often due to ST131[5,8]. Resistance to $\beta$-lactams like 3GCs can lead to increased usage of last-line therapies like carbapenems, with carbapenem resistance in UPEC also associated with ST131[30,31]. In 2017, rates of resistance to fluoroquinolones in ECB cases across Wales were at least 20.3% ($n = 540/2663$)[29]. NICE also promotes the use of nitrofurantoin or trimethoprim (first-line), and pivmecillinam or fosfomycin (second-line) antibiotics against lower UTIs[32]. However, trimethoprim is no longer recommended in Wales for the treatment of UTIs in the 65 and over age group[29]. The increase in antimicrobial-resistant infections is problematic on several levels. For example, patients are more likely to receive inappropriate empirical therapy involving an agent to which the pathogen is resistant. The circulation of strains with extensive levels of resistance to key antimicrobials, such as 3GCs, increases the likelihood of UTI treatment failures, prolonging the length of infection, potentially allowing the bacteria to flourish by removing commensal bacteria which compete for bacterial growth, and increases severe

outcomes such as the risk of a patient developing bacteraemia, resulting in increased morbidity and mortality.

Genomic epidemiology, the use of whole-genome sequencing (WGS) in epidemiological investigations, is increasing worldwide in public health responses. With increasing rates of antimicrobial-resistant ECB, it is vital to understand the genetic relatedness of circulating strains on a local, national, and global scale. This work investigated the evolution of ST131 strains from individuals in Wales with bacteraemia identified over a 12-month period. Genomic sequence data enabled the characterisation of circulating ST131 in Wales, showing multiple introductions of this global clone. Our analyses also reveal multiple bacteraemia cases caused by a unique geographically-restricted, monophyletic subgroup of ST131 within North Wales characterised by ESBL production.

## Results

### *E. coli* ST131 strains from Wales

Previous genomic and multilocus sequence typing (MLST) investigations identified that ECB cases were disproportionately caused by *E. coli* ST131 ($n = 187/720$, 26.0%)[33]. This study involved *E. coli* ST131 isolates cultivated from blood specimens collected from twenty hospitals across six health boards within Wales (Supplementary Materials, Figure S2). Of the 157 isolates collected over the study (between 2013 and 2014), 142 passed quality-control on the sequence data (females $n = 70/142$ (49.3%), males $n = 69/142$ (48.6%), no sex recorded $n = 3/142$ (2.1%)). Patients were typically older, with a median age of 80 years (interquartile range (IQR): 70 to 87 years; range: 19 to 105 years), which reflects the known patient profile of ECB cases[3].

### Major ST131 clades were represented amongst ST131 circulating Wales

The 142 draft genomes had a median total length of 5.20 Mb (IQR: 5.10 to 5.27; range: 4.75 to 5.48 Mb), a median GC content of 50.7% (IQR: 50.7 to 50.8%; range: 50.5 to 51.3%), and a median N50 statistic of 199.59 kb (IQR: 102.39 to 245.60 kb; range: 6.42 to 499.50 kb). All 142 genomes were ST131, except for BA1243 and BA1279 (both likely to be ST131, or same Clonal Complex) which differed in the fumarate hydratase class II (*fumC*) and malate dehydrogenase (*mdh*) genes respectively (Supplementary Materials, Figure S3). To capture a snapshot of the genomic diversity and population structure amongst ST131 strains circulating in Wales, the draft assemblies of the 142 genomes were contextualised with a global collection of ST131 cases sequenced elsewhere ($n = 208$) (Supplementary Materials, Figure S4). The 13,758 non-recombinant core-genome single nucleotide polymorphism (SNP) alignment represents a core-genome alignment of 2,575,140 bp relative to the 5,109,767 bp reference chromosome EC958 (GenBank: HG941718). All three well-supported major ST131 clades (A, B, and C) were represented across Wales. While most isolates were located within clade C ($n = 103/142$, 72.5%), there was no substantial difference between representatives from clade A ($n = 22/142$, 15.5%) and clade B ($n = 17/142$, 12.0%). However, a few genomes - BA1243, BA942, BA2098 from clade A and BA1287, BA1408 from clade B - exceeded the boundaries of the upper and lower 1.5 x interquartile range for genome length (i.e., total genome length <4,841,869 bp and >5,523,239 bp). The Welsh ST131 draft genomes have a high prevalence of chromosomal mutations conferring high-level resistance (MICs >32 mg/L) to fluoroquinolones, where most ($n = 102/142$, 71.8%) contain double variants in *gyrA* (D87N and S83L) and *parC* (E84V & S80I). These 102 genomes were located in clade C. An additional ten Welsh genomes contain a single variant in *gyrA* (S83L), which usually confers low-level resistance (MICs 0.5 mg/L). Of these ten genomes, eight are located in clade A, and two in clade B. The $bla_{CTX-M-15}$ gene (CTX-M-1 group) is the most common ($n = 66/142$, 46.5%) AMR gene encoding ESBLs amongst the 142 Welsh strains. With the exception of two clade A strains (BA130 (SRA: SRR14519479) and BA306 (SRA: SRR14519544),

the $bla_{CTX-M-15}$ gene is confined to clade C2. Only three Welsh genomes in clade C1 carry the $bla_{CTX-M-27}$ gene. The $bla_{OXA-1}$ gene is also common ($n = 62/142$, 43.7%) amongst Welsh strains, which encodes resistance to amoxicillin/clavulanic acid, and can contribute to piperacillin/tazobactam (antibiotic/$\beta$-lactamase inhibitor) resistance amongst ESBL-producing *E. coli*. Notably, 37.3% ($n = 53/142$) of the strains carry both $bla_{CTX-M-15}$ and $bla_{OXA-1}$.

## The majority of clade C ST131 isolates circulating Wales were ESBL-producing, conferring resistance to 3GCs

To infer the phylogenetic relatedness of isolates and determine AMR gene carriage, we created a core-genome SNP alignment of clade C strains only using SPANDx. This alignment of global clade C strains ($n = 238$) comprises 4,142 non-recombinant orthologous biallelic SNPs. These SNPs represent a ~3,890,700 bp core-genome (regions estimated to the nearest 100 bp with ≥95% coverage across all genomes, excluding mobile genetic elements) relative to the 5,109,767 bp chromosome of EC958. Almost half ($n = 102/238$, 42.9%) of our global clade C lineage comprised isolates collected from bacteraemia cases across Wales (Fig. 1; see branch lengths expressed as SNPs in Supplementary Materials, Figure S5). In our combined dataset, the majority of clade C ST131 strains belonged to sub-clade C2 ($n = 161/238$, 67.5%), with sub-clade C1 ($n = 76/238$, 31.9%) less common. Sub-clade C2 mostly comprised Welsh strains ($n = 88/161$, 54.7%), whereas in sub-clade C1, the Welsh strains comprise only 18.4% of the sub-lineage ($n = 14/76$). The majority ($n = 68/102$, 66.7%) of these clade C isolates demonstrate an ESBL-producing genotype. In terms of acquired resistance to $\beta$-lactams, CTX-M-type metallo-$\beta$-lactamase genes were dominant, with the most prevalent being the $bla_{CTX-M-1}$ group ($n = 65/102$, 63.7%). The

second most prevalent $\beta$-lactamase was $bla_{OXA-1}$ ($n = 61/102$, 59.8%), which encodes resistance to amoxicillin/clavulanic acid, and piperacillin/tazobactam (antibiotic/$\beta$-lactamase inhibitor). Among the C2 population (Fig. 1), there is a cluster of 35 isolates from Wales that were separated by a maximum pairwise distance of 112 non-recombinogenic SNPs between strains BA264 (collected in South Wales in 2013) and BA2000 (collected in North Wales in 2014). Strains within this C2 Welsh cluster (designated GB-WLS.C2) were closely related with a median pairwise distance of 46 (IQR: 20 to 87) non-recombinogenic SNPs. The GB-WLS.C2 sub-lineage has descended from a common ancestor (CA) shared with the clinical O25b:H4:K5 strain S125EC (SRA: ERS126605), which was cultivated in 2002 from a patient with a surgical wound in Canada[9], consistent with the global dispersion of ST131.

## The GB-WLS.C2 Welsh cluster reveals unique core-genome mutations and an unprecedented capsule type

Isolates within GB-WLS.C2 were distinguishable by six unique core-genome SNPs relative to EC958, two of which are in genes associated with fitness or virulence (*cusB*; cation efflux system mediating resistance to copper and silver and *fepE*; ferric enterobactin siderophore transport protein) (Table 1). Isolates within GB-WLS.C2 could also be distinguished from other ST131 C2 isolates by their capsule type. GB-WLS.C2 shares a common ancestor with strain S125EC, which was isolated from North America and has a K5 capsule[10]. Instead, however, most strains within GB-WLS.C2 have lost region II of the K5 capsule loci, likely because of recombination, resulting in a K5 variant capsule loci and predicted loss of capsule production (Fig. 2). An additional nine strains across the GB-WLS.C2 phylogeny were typed K- (in silico) and were missing regions I and II of the capsule locus. In *E. coli*, group 2

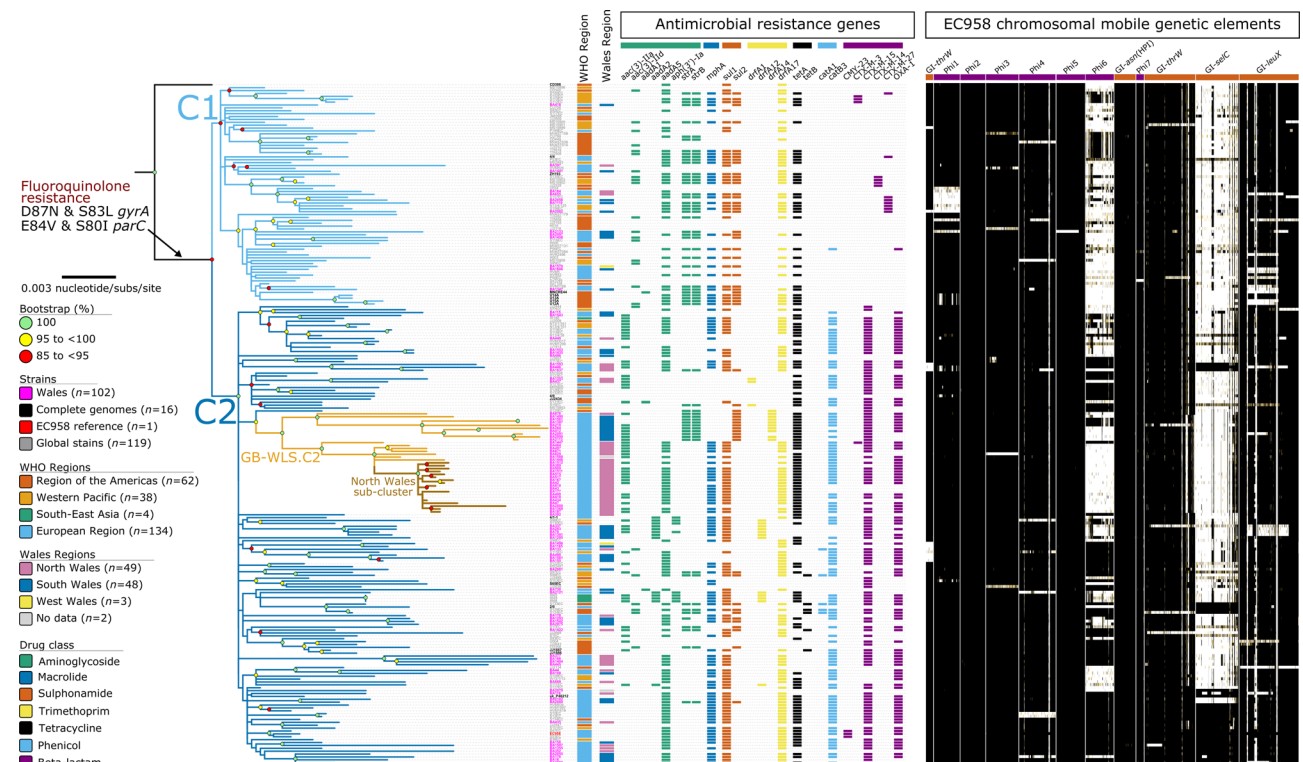

**Fig. 1 | Maximum likelihood phylogeny of clade C *Escherichia coli* sequence type (ST)131 isolates, alongside the antimicrobial resistance genotype and mobile genetic element (MGE) complement.** Phylogeny inferred from 4142 non-recombinant orthologous biallelic core-genome single-nucleotide polymorphisms (SNPs) from 238 strains. Moderate recombination SNP density filtering in SPANDx (excluded regions with ≥3 SNPs in a 100 bp window). SNPs were derived from read mapping to the reference chromosome EC958 (GenBank: HG941718). The

phylogenetic tree is rooted according to the CD306 (GenBank: CP013831) out-group. Branch lengths represent nucleotide substitutions per site as indicated by the scale bar. Bootstrapping using 1,000 replicates demonstrates the robustness of the branches. The presence/absence analysis of loci is based on the uniform coverage at each 100 bp window size in SPANDx. Coverage is shown as a heat map where ≥80% identity is highlighted in black and ≥50% identity is highlighted in yellow. White plots indicate regions that are absent.

**Table 1 | Clade-specific variants identified for GB-WLS.C2**

| Position in EC958 | EC958[a] | GB-WLS.C2 Welsh Cluster[a] | Change[b] | Impact | Codon | Gene | Product |
|---|---|---|---|---|---|---|---|
| 601,912 | **C** | _T_ | Syn | LOW | 383 | _cusB_ | Cation efflux system protein |
| 632,160 | **C** | _T_ | Q = >stop | HIGH | 130 | _fepE_ | Ferric enterobactin transport |
| 1,672,896 | **T** | _C_ | Syn | LOW | 130 | _dosP_ | Oxygen sensor protein |
| 2,698,528 | **G** | _A_ | T = > I | MODERATE | 232 | _fryC_ | Fructose-like permease IIC component |
| 2,902,622 | **T** | _C_ | | | | | |
| 2,964,604 | **A** | _G_ | Syn | LOW | 165 | _pncC_ | Nicotinamide-nucleotide amidohydrolase |

[a]Emboldened and italicised nucleotides are specific to sub-clade C2 EC958 and isolates within the Welsh cluster, respectively.
[b]Consequence of SNP relative to EC958 (C2).
[b]Synonymous change (Syn); non-synonymous changes to protein-coding genes are shown by single letter amino acid abbreviation (EC958 sequence on the left, SNP impact on right); blank lines indicate variant in intergenic region.

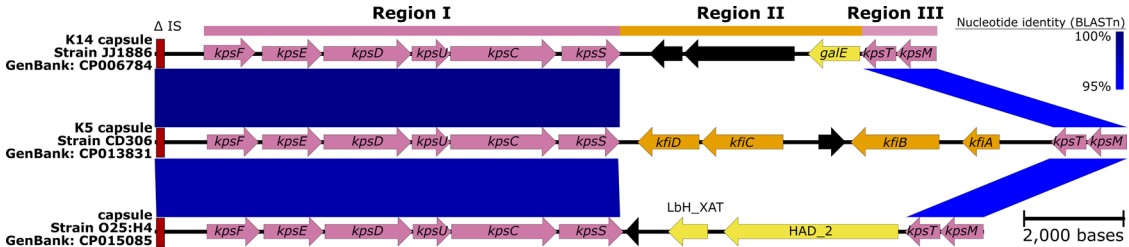

**Fig. 2 | Major structural features and nucleotide pair-wise comparisons of group 2 capsules in _Escherichia coli_ ST131.** Nucleotide comparisons between the sub-clade C2 strain JJ1886, clade C outgroup genome CD306, and clade C2 strain O25:H4, highlighting differences between the K14, K5 and an unprecedented capsular region. The capsule loci in strain O25b:H4 is present in the GB-WLS.C2 draft genomes. Blue shading indicates nucleotide identity between sequences according to BLASTn (95 to 100%). Key genomic regions are indicated: IS: red (185 bp fragment from IS110 family), conserved capsular regions I and III: pink, differing capsular regions (region II): orange and yellow, other CDSs: black. Image created using Easyfig[34].

capsular polysaccharides typically share conserved regions in the capsule loci (regions I and III). These conserved regions encode the transmembrane complex involved in the export and assembly of the capsular polysaccharides[19,35,36]. Region II, however, is serotype-specific and encodes for enzymes responsible for synthesizing the capsular polysaccharide. Strains within GB-WLS.C2 that have the unprecedented capsule type have region II of the capsule locus replaced with two genes, the _catB_ chloramphenicol-related O-acetyltransferase (xenobiotic acyltransferase [XAT]), conferring resistance to chloramphenicol and a HAD-IA family hydrolase, which has been resolved with the genome of the clinical strain O25b:H4 collected in Saudi Arabia in 2014 (GenBank: CP015085). In the C2 strain O25b:H4, the low average guanine-cytosine (GC) content (42.18%) of the 15.2 kb capsule locus as compared to the chromosomal GC content of ~50% suggests that this region was acquired via horizontal gene transfer.

Among GB-WLS.C2, there is a sub-cluster of 18 isolates from North Wales that were separated by a maximum pair-wise distance of 28 (median: 10, IQR: 8 to 13) non-recombinogenic SNPs, highlighting a small, local ST131 cluster. Isolates within this GB-WLS.C2 sub-cluster from North Wales were distinguishable by 10 unique SNPs and a single 1-bp deletion relative to EC958 (Table 2). Additionally, a single strain BA909 (collected in 2014), contained a SNP putatively conferring resistance to rifampicin in _rpoB_ (Q513L)[37]. This analysis identified two strains (BA434 (female aged 80-years) and BA910 (female aged 77-years)) that are separated by a single SNP, from two different hospital laboratories, sampled 84-days apart. This sequence similarity may suggest that these cases were linked by transmission or were colonised/infected from the same source. Additionally, our data from a single hospital laboratory suggests transmission in this same North Wales sub-cluster. For example, isolates BA408 and BA434 from two individual patients collected two days apart from within the same hospital laboratory were

separated by a single SNP, suggestive of a possible epidemiological link.

**Temporal analyses pinpoint the emergence of fluoroquinolone-resistant clade C ST131 to the early 1990s**

The divergence time and evolutionary distance for the 238 clade C ST131 showed a linear relationship (correlation coefficient = 0.62), with the regression analysis in TempEST indicating that the genomes accumulate mutations at a rate of $6.28 \times 10^{-4}$ substitutions per site per year ($R^2 = 0.39$) (Supplementary Materials, Figures S6a and S6b). The time to the most recent common ancestor (MRCA) is estimated at the end of 1991 (95% confidence interval: 1988 to 1994). Likewise, BEAST2 pinpoints the time to MRCA to 1992 (95% highest posterior density (HPD): 1986 to 1998) (Fig. 3) (based on median node height) and estimates a mutation rate of $6.50 \times 10^{-4}$ substitutions per site per year (95% HPD: $5.15 \times 10^{-4}$ to $7.88 \times 10^{-4}$). This translates to 2.7 fixated SNPs per year per genome (95% HPD: 2.1 to 3.3), which means that between four and seven SNPs can be expected to differ between two isolates sharing a MRCA one year prior. This temporal estimate is further supported by a previous study which highlighted the within-host diversity of ST131 residing in the intestinal tract of a single patient[22], with strains U13A (GenBank: CP035477) and U14A (GenBank: CP035516) collected nine months apart. In this study, these two strains were separated by six pairwise SNPs, which based on our estimated mutation rate, is what is expected for isolates with a MRCA of up to one year prior. To correct for ascertainment bias, our dataset describes one SNP for every 965.7 bases across the ~4 Mb core-genome. This translates to a genome-wide mutation rate of $6.73 \times 10^{-7}$ mutations/year/site relative to genome size, which is consistent with previous large-scale temporal analyses of _E. coli_[5] ($4.39 \times 10^{-7}$ [10] and $4.14 \times 10^{-7}$ [13]) and _Shigella_ ($6.0 \times 10^{-7}$ [38]). A Bayesian skyline plot showed a large expansion of ST131 Clade C beginning in the early 2000s before a smaller drop in

**Table 2 | Clade-specific variants for the North Wales sub-cluster**

| Position in EC958 | EC958[a] | Cluster[a] | Change[b] | Impact | Codon | Gene | Product |
|---|---|---|---|---|---|---|---|
| 900,998 | G | A | W = >stop | HIGH | 159 | nfsA | Oxygen-insensitive NADPH nitroreductase |
| 1,052,210 | G | T | G = > C | MODERATE | 403 | ycaQ | Uncharacterized protein |
| 1,113,811 | C | T | Syn | LOW | 74 | appB | Cytochrome bd-II ubiquinol oxidase subunit 2 |
| 1,842,716 | T | C | S = > P | MODERATE | 2 | ydiS | Probable electron transfer flavoprotein-quinone oxidoreductase |
| 2,372,060 | G | A | Syn | LOW | 413 | gatZ | D-tagatose-1,6-bisphosphate aldolase subunit |
| 2,405,414 | CG | C | Deletion | MODIFIER | | | |
| 2,468,793 | G | A | Syn | LOW | 92 | yeiR | Zinc-binding GTPase |
| 2,485,972 | A | G | Syn | LOW | 302 | yejM | Inner membrane protein |
| 3,196,673 | G | A | Syn | LOW | 258 | xanQ | Xanthine permease |
| 4,706,103 | C | G | W = > S | MODERATE | 31 | mdtN | Multidrug resistance protein |
| 4,734,158 | A | G | S = > P | MODERATE | 205 | basS | Sensor protein |

[a]Emboldened and italicised nucleotides are specific to sub-clade C2 EC958 and isolates within the North Wales sub-cluster, respectively
[b]Consequence of SNP relative to EC958 (C2).
[b]Synonymous change (Syn); non-synonymous changes to protein-coding genes are shown by single letter amino acid abbreviation (EC958 sequence on the left, SNP impact on right); blank lines indicate variant in intergenic region

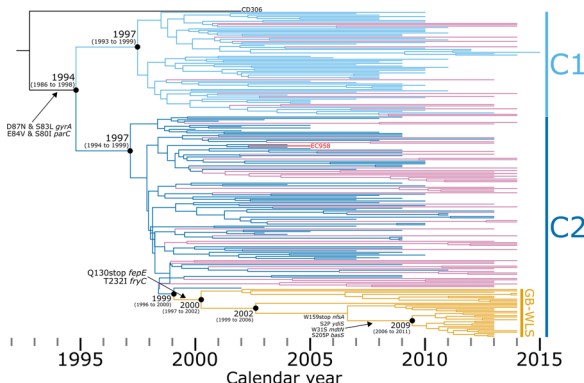

**Fig. 3 | Evolutionary reconstruction of clade C *Escherichia coli* sequence type (ST)131.** A time-calibrated maximum clade credibility tree inferred from 4150 non-recombinant orthologous biallelic core-genome single-nucleotide polymorphisms (SNPs). Moderate recombination SNP density filtering in SPANDx (excluded regions with ≥3 SNPs in a 100 bp window). SNPs were derived from read mapping to the reference chromosome EC958 (GenBank: HG941718). X-axis represents the emergence time estimates. Isolates from Wales are shown with reddish purple or orange branches.

population size during 2007 and subsequent stabilisation from 2010 (Supplementary Materials, Figure S6c).

## Discussion

Recent investigations across England have shown the value of identifying clonal lineages using high-resolution analyses obtainable through WGS for surveillance efforts[5,8]. For example, ECB in England is shown to represent a spillover from strains circulating in the wider human population[5]. This genomic epidemiology study provides a snapshot of the population structure of *E. coli* ST131 associated with bacteraemia circulating in the wider population in Wales. A total of 142 *E. coli* ST131 strains collected across Wales underwent WGS and were contextualised for their genetic relationship with global ST131 strains through available datasets. To the knowledge of the study authors, this represents an initial characterisation of the epidemiological and spatiotemporal nature of ST131 circulating in Wales.

Initial phylogenetic relationships were first inferred from SNPs to depict an overall tree topology that represents core-genome alignments of draft genome assemblies. These initial analyses found that, while all three major clades well-supported by the literature are represented in ST131 strains circulating in Wales, the predominant lineage is defined by a high prevalence of chromosomal mutations conferring resistance to fluoroquinolones and the presence of AMR genes encoding ESBLs (clade C, particularly sub-clade C2). Subsequent analyses were undertaken to compile a much higher resolution of inter-strain relationships by exploiting read mapping of short-read sequence data to a complete genome. The study results highlight the emergence and dissemination of a distinct C2 sub-cluster because of an introduction into Wales circa 2000 (sub-clade C2; 95% HPD: 1997 to 2002), which has been named GB-WLS.C2. These analyses also offered the opportunity to re-evaluate the ST131 clade C evolutionary trajectory previously characterised[5,10,12,13] and confirms the requirement for careful re-analysis of publicly available genomic data, with stringent quality control requirements. For example, a recent investigation highlights the importance of assessing datasets for the presence of mixed strains before phylogenetic analyses[39].

The absence of ST131 genomes from the rest of the UK in this study is acknowledged as a limitation. Although including more ST131 genomes from the rest of the UK could have strengthened our analysis, the study was started in 2017 and relied on available data, notably from a significant ST131 study in 2016[10]. Despite limited publicly accessible UK ST131 genomes (~245), a subset was already incorporated in our research from a follow-up study[40] to Ben Zakour et al.[10]. We acknowledge this limitation's impact on the study's robustness, and exploring additional data sources initially could have bolstered our analysis of within-country transmission dynamics.

The limited genomic diversity (median pair-wise distance of 10 SNPs) amongst a distinct GB-WLS.C2 sub-cluster from North Wales (which emerged in 2009) suggests that the actual reservoir of infection is not confined to a single nosocomial setting. Analyses into local clusters and transmissions can be highly discriminatory, with WGS becoming a routine part of surveillance programs. Further, these data suggested that some individuals with isolates that clustered in GB-WLS.C2 may be colonised/infected from the same source and demonstrate the benefits of incorporating WGS and epidemiological data for public health surveillance.

Of particular concern is the high rates of ESBL carriage (66.7%) in ST131 bacteraemia isolates in Wales (clade C/*H*30), particularly those conferring resistance to 3GCs, from the $bla_{CTX-M-1}$ (*n* = 65/102, 63.7%). These rates are lower than studies in other jurisdictions; 95% of cephalosporin-resistant ST131 isolates in Australia, New Zealand and Singapore showed isolate carriage of $bla_{CTX-M-15}$ or $bla_{CTX-M-27}$[7] and 82.2% of cefotaxime-resistant UPEC isolates from South-West England

carried $bla_{CTX-M}$ variants[41]. However, these study methodologies differed by specifically selecting 3GC-resistant isolates for inclusion. This investigation, instead, is population-based and not biased by AMR selection as samples were sequenced based on their determined phylogenetic groups. The rapid global emergence and sustained dominance of clade C ST131 and the characterisation of the unique ST131 Welsh sub-lineage (GB-WLS.C2) highlights the requirement for timely and continuous annual genomic surveillance, which could facilitate rapid and targeted interventions, for successful infection control, antimicrobial stewardship, and public health response. The Office for National Statistics has previously collated mortality data where *E. coli* septicaemia or sepsis were explicitly mentioned on death certificates in Wales between 2001 and 2015[3]. Gwasanaeth Iechyd Gwladol (GIG) Cymru, or the National Health Service (NHS) in Wales, can substantially benefit from collating this mortality data with genomic surveillance. This collaborative approach would ensure the timely provision of data, facilitating swift actions in response to potentially life-threatening infections from both healthcare- and community-associated origins, with little delay.

This study estimates the emergence of fluoroquinolone-resistant C ST131 circa 1994 (95% HPD: 1986 to 1998). This differs from the previously reported dates from Ben Zakour et al.[10], Stoesser et al.[12], and Kallonen et al.[5], which estimate 1987 (95% HPD: 1983 to 1992), 1982 (95% HPD: 1948 to 1995), and circa 1987, respectively. In contrast, after analysing 794 ST131 genomes, Ludden et al.[13]. is closer to our findings and pinpoints the emergence of the fluoroquinolone-resistant C ancestor to 1992 (95% HPD 1989 to 1994). While the posterior mean/median node heights for the clades vary between studies, it is important to recognise that the 95% HPD intervals overlap. The variation in study results may be due to the updated methodology utilised, including: stringent quality control metrics, newer versions of tools and methods, use of a high-quality clade C reference genome, the exclusion of clade B ST131 strains (which may interfere with the temporal signal), and the inclusion of a fluoroquinolone sensitive clade C outgroup strain (CD306). However, we cannot overlook that these differences could also be due to differences in the Bayesian approaches used in each study[42,43].

These highly discriminatory analyses reveal multiple introductions of sub-clade C2 into Wales before an emergence circa 2000 (95% HPD: 1997 to 2002) of the unique clonal sub-lineage (GB-WLS.C2), which shares a CA with a genome collected from North America. However, it is crucial to note that this study lacks direct transmission analyses using ST131 genomes from the rest of the UK. As a result, while these analyses provide insights into the genomic relationships, the claim of multiple introductions should be considered cautiously without specific transmission analyses involving UK isolates. These unique strains (GB-WLS.C2) were related with a median pair-wise SNP distance of 46 non-recombinogenic SNPs, which could indicate localised transmission with an unidentified infection reservoir. The CA to this unique strain (GB-WLS.C2) is distinguishable from that shared with the basal S125EC strain by an impairment of ferric enterobactin synthesis and transport due to a premature termination because of a Q130stop codon in *fepE* and an unprecedented capsule-type. While there is a multiplicity of K antigens across ST131 Clade C2, GB-WLS.C2 strains differ from the CA shared with strain S125EC by a switch from a K5 to an unprecedented capsule antigen due to a recombination event at Region II of the capsular locus. This unprecedented capsule type is also present in the ST131 Clade C2 strain O25b:H4 (GenBank: CP015085) from Saudi Arabia, however this strain does not cluster with GB-WLS.C2 isolates. Notably, both enterobactin and the capsule are known UPEC virulence factors. Whether the predicted inactivation of these loci resulted in a decrease in virulence remains to be elucidated and represents a research question for future investigation. For strains within GB-WLS.C2, region II of the capsular loci was replaced with a chloramphenicol acetyltransferase (CAT) and HAD-IA

family hydrolase. CATs inactivate chloramphenicol by generating derivatives like 1-acetoxy chloramphenicol, 3-acetoxy chloramphenicol, or 1,3-diacetoxy chloramphenicol. These derivatives are unable to inhibit bacterial growth and survival as interruption of the ribosomal peptidyl-transferase is no longer possible[44,45]. Further, the identification of a local cluster within North Wales, MRCA emerged in 2009, with very closely related strains differing by a median of 10 non-recombinant pair-wise SNPs, suggests that there was possible direct transmission between these individuals. Although, the anonymisation of patient data limits confirmation of an actual nosocomial infection reservoir and evidence of negative (or positive) culture on admission would be required for any certainty.

The study design describes cases of bacteraemia, with confirmed blood cultures, caused by ST131 in Wales. This is likely a consequence of UTI treatment failure due to AMR, although further research is needed to establish links between confirmed blood and urine isolates. Therefore, this study may not represent the whole population structure of GB-WLS.C2 in UTIs in Wales. In this circumstance, one would expect increased rates of AMR, and thus it is likely that the population structure of all GB-WLS.C2 is to be less resistant than might be expected based on the results reported here. Globally there is a necessity to acquire a deeper understanding of the population structure of UPEC, so that UPEC strains that are more likely to result in treatment failure and progress to bacteraemia can be identified as a risk factor. This can be achieved by identifying and tracking genomic sequences (e.g., AMR determinants and virulence factors) as indicators for predicting phenotypic characteristics. One of the key strengths of our study was our ability to avoid a temporal or geographical bias in our dataset by contextualising the ST131 Welsh strains with global isolates. However, while including more ST131 genomes collected from the rest of the UK could potentially introduce temporal variations, the aim was to ensure a comprehensive representation for a more accurate understanding of broader evolutionary relationships of ST131. Although more genomes might introduce some temporal variations, geographically diverse samples contribute to a broader representation of the global ST131 population, potentially enhancing the reliability of our findings rather than creating bias. This lack of bias was reflected by our population expansion timeline, which coincides with the first reported isolate of ST131 in the UK in 2003[5], before becoming the predominant clone ($n = 52/88$, 59.1%) in the Northwest of England (albeit mostly urine-derived strains) between 2004 and 2006[8]. Despite this, expanding the dataset by including additional UK isolates would have indeed increased the breadth of our analysis, aiding in a more comprehensive understanding of within-country transmission dynamics.

Genomic epidemiological analyses on 142 ST131 strains associated with bacteraemia across Wales between 2013 and 2014 were performed using whole-genome sequencing. This research demonstrates how reanalysis of publicly available genomic data, with stringent quality control, can improve on the evolutionary trajectory of the ST131 clade C previously characterised. Additionally, this study showed geographical clustering of sub-clade C2 in North Wales; characterised by genotypic resistance to third-generation cephalosporins, fluoroquinolones, chloramphenicol, and nitrofurantoin. This follows the introduction of a single sub-lineage into Wales circa 2000 and its expansion and persistence, which has been named GB-WLS.C2. This study also displays a localised cluster of ST131 bacteraemia in Wales captured between 2013 and 2014. Further, this study highlights the need to incorporate whole-genome sequencing with epidemiological data to identify potential infection reservoirs in the environment, which will allow for identifying ST131 transmission dynamics between healthcare settings and the community. By gaining a detailed understanding of significant *E. coli* bacteraemia strains, it should be possible to develop targeted public health measures to reduce the risk of *E. coli* bacteraemia and act to combat the rise of antimicrobial resistance.

## Methods

### Welsh *E. coli* isolate collection and genome sequencing

Microbiology laboratories in Public Health Wales (PHW) and across the Welsh NHS were asked to submit all *E. coli* blood isolates from blood samples collected between April 2013 and March 2014, to the national Specialist Antimicrobial Chemotherapy Unit (SACU) at the University Hospital of Wales (also known as Ysbyty Athrofaol Cymru). The isolate dataset was linked to routine microbiological surveillance data by PHW to obtain isolate AMR profiles. AMR profiles were characterised by polymerase chain reaction (PCR). Selected samples were sequenced based on their determined phylogenetic groups. Isolates were transported to Cardiff University, cultured overnight in liquid culture, and extracted using a Promega (Wisconsin, Unites States) Maxwell instrument. Samples were sequenced as paired-end reads on either the NextSeq 500 or HiSeq 2500 platform (Illumina Inc, San Diego, California, United States) at the Oxford Genomics Centre (https://www.well.ox.ac.uk/ogc/) or MicrobesNG (https://microbesng.com/, accessed 22 January 2024). DNA libraries were prepared using a mixture of the Nextera XT® Library Preparation Kit and the NEBNext® Ultra™ Library Preparation Kit (Illumina Inc, San Diego, California, United States), following the manufacturer's instructions in both cases.

This genomic surveillance of ECB in Wales collected 157 non-duplicate clinical *E. coli* ST131 strains as part of enhanced micro-biological surveillance data from hospitals across six administrative units known as health boards (Supplementary Materials, Figure S2). Patient anonymity was maintained by pseudonymised data that went outside PHW. Epidemiological information, including isolate names and available metadata are summarised in Supplementary Data 1 and 2. The WGS of the 157 *E. coli* isolates generated a median of 0.89 million paired-end reads per sample (IQR: 0.43 to 1.09 million; range: 0.14 to 3.37 million) (Supplementary Data 2). Sequence read data for all Welsh isolates were submitted to the National Center for Biotechnology Information (NCBI) Sequence Read Archive (SRA) under BioProject accession number PRJNA729115. The methods used for quality control for this dataset are available in the Supplementary Methods. Briefly, we identified and excluded the sequence data for 15 isolates from further analysis based on the sequencing coverage below 20-fold (Supplementary Data 3).

### Draft genome assembly

Quality-trimmed paired-end reads for the remaining 142 Welsh strains were de novo assembled using MGAP (https://github.com/dsarov/MGAP---Microbial-Genome-Assembler-Pipeline, accessed 22 January 2024), which implements: Velvet v1.2.10;[46] VelvetOptimiser (https://github.com/tseemann/VelvetOptimiser, accessed 22 January 2024); GapFiller v1.10;[47] ABACAS v1.3.1[48] (scaffolds against the chromosome of *E. coli* ST131 strain EC958 (GenBank: HG941718 [https://www.ncbi.nlm.nih.gov/nuccore/HG941718])); IMAGE v2.4;[49] SSPACE v2.0;[50] Pilon v1.22;[51] and MIRA v4[52]. Contigs from the draft assemblies were ordered against the complete chromosome of EC958 using Mauve version snapshot_2015-02-25[53]. QUAST v4.5[54] assessed the assembly statistics generated from MGAP by comparing each isolate to EC958 (Supplementary Data 4).

### Complementary datasets

To facilitate the geographic analysis of the 142 Welsh ST131 isolates within the global context, available genome sequence data were downloaded including: (i) sequence read data from the NCBI SRA using the 'prefetch' and 'fastq-dump' tools within the SRA Toolkit v2.9.0-mac64 (http://ncbi.github.io/sra-tools, accessed 22 January 2024); (ii) draft assemblies; and (iii) associated metadata from Ben Zakour et al[10]. (*n* = 189) and Kidsley et al[40]. (*n* = 19). Notably, six draft assemblies from the Ben Zakour et al. study were replaced with the complete chromosomes: CD306 (GenBank: CP013831); JJ1886 (GenBank: CP006784); JJ1887 (GenBank: CP014316); JJ2434 (GenBank: CP013835); S65EC

(GenBank: CP036245); and ZH193 (GenBank: CP014497) (Supplementary Data 5). The methods used for in silico gene typing and generation of an assembly-based ST131 phylogeny (initial context) for this global dataset are available in the Supplementary Methods.

### Compiling a high-quality ST131 clade C global dataset and identifying genetic variants

For context, the Welsh clade C strains (*n* = 102) were analysed alongside a global collection of clade C ST131 strains (*n* = 117) from three published studies[9–11] as featured in Kidsley et al.[40]. Several complete genomes were integrated by simulating error-free reads using ART (version ART-MountRainier-2016-06-05)[55] to 60x coverage with an insert size of 340 ± 40 bp. These included known clade C ST131 genomes: 2/0 (GenBank: CP023853); 4/0 (GenBank: CP023849); 4/4 (GenBank: CP023826); 4/1-1 (GenBank: CP023844); MNCRE44 (GenBank: CP010876); U12A (GenBank: CP035476); U13A (GenBank: CP035477); U14A (GenBank: CP035516); U15A (GenBank: CP035720); and uk_P46212 (GenBank: CP013658). For this investigation, clade C strains JJ2183 (SRA: SRS456889), MVAST0036 (SRA: SRS456851), MVAST046 (SRA: SRS456881), and MVAST077 (SRA: SRS456882) were removed from the dataset based on the average sequence coverage depth below 20-fold. High-resolution analyses of genetic variants was performed using the Burrows–Wheeler Aligner v0.7.15;[56] SAMtools v1.2;[57] Picard v2.7.1 (https://github.com/broadinstitute/picard, accessed 22 January 2024); the Genome Analysis Tool Kit v3.2-2 (GATK);[58,59] BEDTools v2.18.2;[60] and SNPEff v4.1[61] as implemented in SPANDx v3.2[62]. In brief, the trimmed reads were mapped to the complete chromosome of EC958, which was isolated in March 2005 in the United Kingdom from a community-onset urine infection in an 8-year-old girl[15]. Our final dataset consisted of 245 genomes representing previously published datasets (*n* = 127, including 16 complete genomes) and our Welsh collection (*n* = 102) (Supplementary Data 6).

### High-resolution phylogeny of the clade C ST131 sub-lineage

The 245 strains (EC958 reference (*n* = 1) and dataset (*n* = 244)) were assessed for the presence of strain mixtures. Briefly, this approach flagged seven strains as a probable mixture based on the high number of ambiguous SNPs (from a total of 7,592 SNPs) when compared with the remaining 238 genomes (Median 51 (0.7%); IQR 28 to 104 (0.4 to 1.4%); range 3 to 191 (0.0 to 2.5%)) (see Supplementary Methods). The quality-trimmed paired-end Illumina reads from the remaining 238 high-quality clade C isolates were mapped onto the chromosome of EC958 as described above. SNPs within regions of high-density clusters (≥3 SNPs found within a 100 bp window) and predicted recombination sites (identified using Gubbins v2.4.1[63]) were removed from the core-genome alignment. Sites were excluded if a SNP was called in regions with less than half or greater than 3-fold the average genome coverage on a genome-by-genome basis. The generated alignment consisted of 4142 non-recombinant, orthologous, biallelic core-genome SNPs from the 238 strains. Lastly, a maximum likelihood (ML) phylogenetic tree from the non-recombinant SNP alignment was generated using RAxML v8.2.10[64] (GTR-GAMMA correction) thorough optimisation of 20 distinct, randomised maximum parsimony trees, before adding 1000 bootstrap replicates. The resulting phylogenetic tree was visualised using FigTree v1.4.4 (http://tree.bio.ed.ac.uk/software/figtree/, accessed 22 January 2024) and EvolView v2[65,66].

### Temporal analysis

A regression analysis was used to estimate the temporal signal in the clade C ST131 sub-lineage between the root-to-tip genetic distance using TempEst v1.5.15[67]. The ML phylogenetic tree that was reconstructed from the alignment of 4142 non-recombinant, orthologous, biallelic core-genome SNPs (as described above) was used as the input into TempEST. To further the temporal analysis, a time-calibrated phylogenetic tree was generated with BEAST2 v2.6.1[68]. The alignment

of 4142 SNPs was run through jModelTest v2.1.10[69,70], which identified the GTR nucleotide substitution model as the best-fit evolutionary model. To test if the strict clock or uncorrelated relaxed clock best-fit our dataset, initial models were created using tip dates, a GTR substitution model, and a coalescent prior with a constant population. Both models were tested with the Nested sampling Bayesian computation algorithm v1.1.0 within the BEAST2 package with a particle count of 1, sub chain length of 5000, and Epsilon of $1.0 \times 10^{-6}$. This analysis provides evidence in favour of the uncorrelated relaxed clock model. Various population models were compared to ensure selection of the best-fit model. These included the Bayesian skyline, coalescent constant, and exponential growth population size change models. The Gamma Site Model Category Count was set to four and the GTR substitution model rates determined from jModelTest were included (i.e., rate AC = 0.93, AG = 3.13, AT = 1.12, CG = 0.13, CT = 3.08, and GT = 1.00). Notably, the initial clock rate was set to $6.28 \times 10^{-4}$ as estimated from the root-to-tip regression analysis in TempEST) with a uniform distribution and an upper bound of 0.1. All other priors were left as default. A total of three independent Markov chain Monte Carlo (MCMC) generations for each analysis were conducted for 100 million generations. Trees were sampled every 1,000 generations which resulted in triplicate samples of 100,000 trees for each model test. All BEAST runs were imported into Tracer v1.7.1 (http://github.com/beast-dev/tracer/, accessed 22 January 2024) to assess statistics. LogCombiner v2.5.0 (BEAST 2 package) then combined the replicated analyses for each model with a 10% burn-in to assess convergence/appropriate sampled run. Finally, TreeAnnotator v2.4.5 (BEAST 2 package) removed the 10% burn-in and generated maximum clade credibility (MCC) trees for each run (established from 243 million trees), reporting median values with a posterior probability limit set at 0.5. FigTree was used to visualise the annotated MCC trees. We determined the best-fitting tree model as the uncorrelated relaxed exponential clock model with the Bayesian skyline population size change model based on the mean tree likelihood scores (Supplementary Data 7).

### Reporting summary

Further information on research design is available in the Nature Portfolio Reporting Summary linked to this article.

## Data availability

The study sequences are available in the National Center for Biotechnology Information (NCBI) under BioProject accession number PRJNA729115. Raw Illumina sequence read data have been deposited to the NCBI sequence read archive (SRA (https://www.ncbi.nlm.nih.gov/sra)) under the accession numbers SRR14519411 to SRR14519567 [https://www.ncbi.nlm.nih.gov/bioproject/?term=PRJNA729115]. A complete list of SRA accession numbers is available in Supplementary Data 1 (available in the online version of this article). The high-quality draft assemblies have been deposited to GenBank under the accession numbers JAHBGJ000000000 to JAHBMG000000000, and JAHBRR000000000 to JAHBRT000000000 [https://www.ncbi.nlm.nih.gov/bioproject/?term=PRJNA729115]. Publicly available genome sequence data downloaded for comparative analyses are available in NICU under BioProject accession numbers: PRJDA19053; PRJEA61443; PRJEB2968; PRJNA211153; PRJNA218163; PRJNA307507; PRJNA311313; and PRJNA627752. The programs used to analyse raw sequence reads for polymorphism discovery and whole-genome sequencing based phylogenetic reconstruction are available as described in the materials and methods. The authors confirm all supporting data and protocols have been provided within the article or through supplementary data files.

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

## Acknowledgements

We thank the Microbiology laboratories in Public Health Wales and across the Welsh NHS, and the Specialist Antimicrobial Chemotherapy Unit (SACU) at the University Hospital of Wales for contributing samples to the national *E. coli* bacteraemia project undertaken by the Healthcare Associated Infection, Antimicrobial Resistance & Prescribing Programme (HARP). HARP provided project management/epidemiological expertise for the study methodology, including dataset preparation, sample selection and transportation. We acknowledge the facilities and the scientific and technical assistance of staff at the Oxford Genomics Centre and MicrobesNG at the University of Birmingham. This research was supported by QRIScloud and by use of the Nectar Research Cloud. The Nectar Research Cloud is a collaborative Australian research platform supported by the National Collaborative Research Infrastructure Strategy (NCRIS). Sequence data are uploaded and stored on the centralised Cloud Infrastructure for Microbial Bioinformatics (MRC-CLIMB) server. This work was funded by the Medical Research Council (grants MR/L015080/1 to TRC and MB and MR/T030062/1 to TRC), which provided staff time and the key computational resources for data storage and analysis, and the Wellcome Trust (project funding via the Cardiff University ISSF fund) which funded sequencing and initial analysis of the samples (to TRC, CB and MB). The author would like to thank Derek Sarovich and Erin Price (Centre for Bioinnovation at the University of the Sunshine Coast, and the Sunshine Coast Health Institute) and Thomas Cuddihy (QFAB Bioinformatics and Research Computing Centre, The University of Queensland) for high-performance computing support and helpful discussions about software functionality. For the purpose of open access, the author has applied a CC BY public copyright licence.

## Author contributions

Conceptualisation: R.T.W. Investigation: R.T.W. Funding was acquired by T.R.C. and computational resources were supported by S.A.B. Formal analysis: R.T.W. Wet-lab experiments: M.J.B. and C.R.B. Data analysis: R.T.W. Dataset preparation/curation: Public Health Wales' Healthcare Associated Infection, Antimicrobial Resistance & Prescribing Programme (HARP) for the national *E. coli* bacteraemia project's whole-genome sequencing work package in collaboration with the Specialist Antimicrobial Chemotherapy Unit (SACU) at the University Hospital of Wales, J.M.A., M.W., T.R.C., and R.T.W. Illumina sequencing was done at the Oxford Genomics Centre and MicrobesNG at the University of Birmingham. Supervision: T.R.C., B.M.F., and S.A.B. Writing (Original Draft Preparation): R.T.W. Writing (Review and Editing): R.T.W., M.J.B., C.R.B., J.M.A., M.W., L.S.J., R.A.H., M.M., M.M.A., B.M.F., T.R.C., and S.A.B. All authors have read and approved the final version of the manuscript.

## Competing interests

The authors declare no competing interests.

## Ethical approval

This work was carried out as part of a larger project examining *E. coli* bacteraemia in Wales. The National Institute for Social Care and Health Research (NISCHR) research ethics committee was approached with an outline of the project to determine whether ethical agreements would be required for the work to be undertaken. The view of the committee was that whilst they would classify the project as research, sit was research that fell within the remit of Public Health and was exempt from requiring ethical approval. This work was undertaken on stored bacterial cultures and no additional clinical samples were collected from any persons to facilitate this study. Patient anonymity was ensured by Public Health Wales' *E. coli* bacteraemia project manager by preparing a pseudonymised study dataset and samples for whole-genome sequencing and analysis by Cardiff University.
