## [Peer Review File · Nature Communications]

Genomic epidemiology reveals geographical clustering of multidrug-resistant *Escherichia coli* ST131 associated with bacteraemia in WalesREVIEWER COMMENTS

Reviewer #1 (Remarks to the Author):

Manuscript by White et al. describes WGS analysis of the main multi-drug resistant clonal group of bacteremia-causing *E. coli* - the clonal group ST131, subclade C/fimH30, isolated in the Wales region of UK. This clonal group is globally spread and, currently, appears to be the most successful and lethal group of extra-intestinal *E. coli*. Thus, better understanding of its epidemiology, resistance mechanisms and pathogenesis is highly warranted.

The study is very well done and uses some of the most advanced bioinformatics tools. Identification of the Wales endemic sub-clone is interesting, especially considering that it is both ESBL producer and resistant to nitrofurantoin – the first recommended antibiotic for UTIs. If this subclone is indeed highly fit, it has potential to spread beyond Wales. Finding the basis of its superior fitness could be of great interest.

Overall, this study illustrates well the potential power of WGS in surveillance of multi-drug resistant strains, which could help with containing the resistance spread. All this said, however, finding that a particular group of strains is endemic to a specific geographic region is not novel, especially considering that these are likely to be hospital-associated strains that could spread between the hospitals in the area.

Reviewer #2 (Remarks to the Author):

This manuscript presents a detailed genomic study of 142 NGS sequenced *E. coli* ST131 blood isolates from Wales, UK. The isolates were collected during 2013-2014. The collection is rather small when compared to some other studies from the UK, but this is the first study to include isolates from Wales. This is the only original contribution of the manuscript. The bioinformatic analyses are detailed and methodologically solid. The Welsh strains are also put into a global context in the bioinformatic analyses. ST131 is one of the most important bacteremia and UTI causing *E. coli* ST and it is also were resistant to antibiotics which emphasises the importance of this research.

Major comments

The study claims that a new clade identified in Wales has the CA in North America, but there is no UK isolates in the reference material? If not, it is not reasonable to hypothesise that the GB-WLS.C2 clade would originate from North America. There are a large number of UK strains publicly available and they should be included in the study to rule out a within UK transmission and origin of the clade. These should be included in the analysis for the authors to make this claim. If this is not possible, the claim should be removed from the manuscript and this limitation should be made clear. In general, it is baffling why UK isolated were not included in the reference dataset for the geographic comparison. I think this makes the manuscript a lot weaker than it would otherwise be.

I have serious concerns that the samples were not sequenced deep enough. The genomic coverage based on the number of reads is very low. There is clear correlation with the number of reads and size of the assembled genome indicating that the sequencing coverage was not high enough. The authors should e.g. clarify how good the coverage across the genome to assure the reader that this was enough for a reliable analysis otherwise the results cannot be trusted.

Minor comments

Line(s)

136-138 While describing the three clades of ST131 it should be noted that clade A is of a different serotype than B and C. This should be added to the text.

139-140 Clade C is divided into three sublineages C(0), C1 and C2 e.g. by Ben Zakour et al 2016. C0 being susceptible to fluoroquinolones.

143-148 Results from McNally et al 2019 should be added to the list e.g. ST131 has an open pangenome and low genomic fluidity which is rare in E. coli and the diversification of certain genetic loci working under the NDFS model.

153-157 Claiming that a guideline recommending the use of betalactams, trimethoprim, cipro or amoxiclav for the treatment of pyelonephritis introduced in 2018 could be contributing to the increasing rates of ESBL-producing E. coli in the UK and referring to an article from 2007 (Livermore et al) is not right.

180 localised and national transmission? With what reference material this claim can be made?

191 profiles with phylogeography???

207-208 (and Table S2) Median doesn't illustrate the situation very well since the extremities are very far from each other. It is good that the range is presented since 0.14 million reads is only 9x coverage. It is good that all isolates under 20x coverage were excluded. It might be worthwhile to present the read counts for the isolates used in the actual analyses.

224 Combine Tables S2 and S4. All the stats from the genomes should be in the same table. Now it's difficult to assess the data. There is clear correlation between the N50 and number of contigs as well as N50 and genome size. I think this is an indication of the low coverage and low number of reads, but now it's difficult to compare these numbers to the ones above. When I combined the data myself, it is quite clear that there is a strong correlation with low number of reads and small genome size indicating that the sequencing coverage is too low.

228 What does "available datasets were downloaded" mean? There are most likely more than 1000 sequenced ST131 genomes available since McNally et al used almost 900 publicly available genomes in their analyses published in 2019 (BioRxiv 2018).

235 Why use a global dataset only? There are a large number of sequenced ST131 isolates from the UK. At least some of them should be included in the analysis. Also why include ST131 isolates from animals in Australia? How do these isolates contextualise the Welsh blood isolates? With this as a reference material in the geographic analysis all within UK links are missed, which, I feel, would be the most important ones. Also, in Table S5 presenting this data, I think it should read *Canis lupus familiaris* not *Canus lumpus familiaris*. Also, some species names e.g., chicken, are not in Latin. Why?

308-309 Is this only based on the tree likelihoods? The model should be selected based on the stepping-stone or path sampling analyses.

326 Which STs were the two non-ST131 isolates? Please add to the text.

445 This should be discussed with the results from Brodrick et al. 2017 which has data on within-patient diversity.

450-451 It's difficult to differentiate the reference from the number. Also, Kallonen et al 2017 should be referred here.

502-503 Since this study is more similar in approach to Kallonen et al 2017, the prevalence on CTX-M-15 and other antibiotic resistance genes could be compared to the numbers from that paper.

527-528 I agree that this study includes highly discriminatory analyses, but no transmission analyses with UK material. Therefore, claiming multiple introductions is too strong a claim.

562-563 How would inclusion of more isolated from the UK introduce bias to the analyses if you

are looking for common ancestors? Temporal maybe, but geographical? In any case, all studies are analysing a small part of the actual bacterial population and including more strains should increase the reliability of the results, not create a bias.

565 Even though Kallonen et al. 2017 reported the first ST131 isolate in the UK, that definitely is not the first. They couldn't have been that lucky. Initial detection is therefore too strong a wording. Maybe first reported isolate would be better?

566 It should be added to the text that Lau et al 2008 were mostly urine isolates.

577-578 Claims "Further, this study highlights the need to incorporate whole-genome sequencing with epidemiological data and a 'One Health' approach to identify potential infection reservoirs in the environment". Where does this study highlight the need for a One Health approach? I do not see it anywhere. There are some studies from the UK that would indicate that a One Health approach is not needed e.g. Ludden et al 2019 mBio. I think that this claim should be removed from the text. I agree with incorporating WGS and epidemiological data, but not One Health.

Table S2

There is something wrong with the data. For example, on row 146. Under Species #2 is Enterobacteriaceae bacterium0.04. Under No. reads Species #3 (%) it reads Salmonella (285) and under Species #3 it says enterica. Please go through the data to make sure everything is correct.

Reviewer #3 (Remarks to the Author):

White and colleagues report on the genomic epidemiology of UPEC ST131 associated with bacteraemia in Wales between 2013 and 2014. The authors used whole-genome sequencing to characterize circulating ST131 strains and determine their genetic relatedness to previously reported sequence data in Wales and other parts of the world. The objective of the cross-sectional population-based study was clearly expressed and an appropriate methodology, which was explicitly presented to allow for reproducibility, was used. Data analyses and interpretation were properly presented in a consistent manner and the conclusion was well supported by the presented data. The manuscript provides comprehensive valuable data on the evolutionary pattern, geographical clustering and transmission source of circulating ST131 strains which are of public health importance in the prevention of E.coli associated bacteraemia as well as antimicrobial resistance.

Major comments

1. The study duration was 12 months which directly impacts the number of E.coli blood isolates collected from PHW and Welsh NHS. Did the authors calculate the sample size before collection, and was the study period and sample size adequate for a good representation of the general circulating strains in Wales?
2. The study limitations were not highlighted in the manuscript.

Minor comment

Line 334, the frequencies of clade A (15.5%) and clade B (12.0%) are not similar. It would be more appropriate to state there was no significant difference between both.

REVIEWER COMMENTS

Reviewer #1 (Remarks to the Author):

Manuscript by White *et al.* describes WGS analysis of the main multi-drug resistant clonal group of bacteremia-causing *E. coli* - the clonal group ST131, subclade C/*fimH30*, isolated in the Wales region of UK. This clonal group is globally spread and, currently, appears to be the most successful and lethal group of extra-intestinal *E. coli*. Thus, better understanding of its epidemiology, resistance mechanisms and pathogenesis is highly warranted.

The study is very well done and uses some of the most advanced bioinformatics tools. Identification of the Wales endemic sub-clone is interesting, especially considering that it is both ESBL producer and resistant to nitrofurantoin – the first recommended antibiotic for UTIs. If this subclone is indeed highly fit, it has potential to spread beyond Wales. Finding the basis of its superior fitness could be of great interest.

Overall, this study illustrates well the potential power of WGS in surveillance of multi-drug resistant strains, which could help with containing the resistance spread. All this said, however, finding that a particular group of strains is endemic to a specific geographic region is not novel, especially considering that this are likely to be hospital-associated strains that could spread between the hospitals in the area.

Author's response: We appreciate the reviewer's positive feedback on our paper and have addressed all comments, as suggested. The reviewer's constructive critique regarding the endemic strains in specific geographic regions, particularly those linked to hospitals and their local spread, is duly noted. We would like to take this opportunity to mention that between April 2013 and March 2014, whole-genome sequencing of *E. coli* from blood isolates was not a standard procedure. This is the first study to include genomes from Wales. While identifying an endemic group of strains isn't ground-breaking. We consider the sequence data and the forthcoming publication to be beneficial for continuous sequencing efforts involving *E. coli* ST131 in Wales, and the rest of the United Kingdom.

Reviewer #2 (Remarks to the Author):

This manuscript presents a detailed genomic study of 142 NGS sequenced *E. coli* ST131 blood isolates from Wales, UK. The isolates were collected during 2013-2014. The collection is rather small when compared to some other studies from the UK, but this is the first study to include isolates from Wales. This is the only original contribution of the manuscript. The bioinformatic analyses are detailed and methodologically solid. The Welsh strains are also put into a global context in the bioinformatic analyses. ST131 is one of the most important bacteremia and UTI causing *E. coli* ST and it is also were resistant to antibiotics which emphasises the importance of this research.

Author's response: We thank the reviewer for the positive and very constructive comments about our paper. We have addressed all comments as suggested and to the best of our ability.

Major comments

The study claims that a new clade identified in Wales has the CA in North America, but there is no UK isolates in the reference material? If not, it is not reasonable to hypothesise that the GB-WLS.C2 clade would originate from North America. There are a large number of UK strains publicly available and they should be included in the study to rule out a within UK transmission and origin of the clade. These should be included in the analysis for the authors to make this claim. If this is not possible, the claim should be removed from the manuscript and this limitation should be made clear. In general, it is baffling why UK isolated were not included in the reference dataset for the geographic comparison. I think this makes the manuscript a lot weaker than it would otherwise be.

Author's Response: We sincerely appreciate the insightful critique provided by the reviewer regarding the limitations of our study, specifically concerning the absence of UK isolates in our reference dataset and its implications for our conclusions. We have amended the text and removed the hypothesise that the GB-WLS.C2 clade would originate from North America; as follows: "These highly discriminatory analyses reveal multiple introductions of sub-clade C2 into Wales before an emergence circa 2000 (95% HPD: 1997 to 2002) of the unique clonal sub-lineage (GB WLS.C2), which shares a CA with a genome collected from North America".

We recognise that including more UK isolates would have significantly enhanced the robustness of our analysis by allowing for a more comprehensive assessment of within-country transmission dynamics. However, this study began in 2017, so we were aligning the genomes from Wales with the significant ST131 study conducted by Ben Zakour *et al.* in 2016. Looking back, an alternative option could have been to retrieve genomes from the Enterobase website. However, at the onset of our study, only ~245 *E. coli* ST131 genomes from the United Kingdom were accessible publicly. I've had a look in the dataset and among these, 27 were already included in Ben Zakour *et al.* 2016 and subsequently in our research. Expanding the number of genomes included in this paper would require a substantial effort and potentially restarting the study from scratch. We regret not exploring these options in our original research.

Reviewer comment: I have serious concerns that the samples were not sequenced deep enough. The genomic coverage based on the number of reads is very low. There is clear correlation with the number of reads and size of the assembled genome indicating that the sequencing coverage was not

high enough. The authors should e.g. clarify how good the coverage across the genome to assure the reader that this was enough for a reliable analysis otherwise the results cannot be trusted.

Author's Response: We thank the reviewer for raising this concern. In our workflow, we assessed the read metrics for 157 Welsh isolates (Table S2) before assembly. This identified 15 strains that were removed from further analyses (Table S3). The 142 remaining strains were then *de novo* assembled (Table S4). You will note that “The 142 draft genomes had a median total length of 5.20 Mb (IQR: 5.10 to 5.27; range: 4.75 to 5.48 Mb)”. Based on the reviewer's comment, I have expanded this a bit further by checking whether the total assembled genome length for any strains fell outside the upper (Q3) and lower (Q1) 1.5 x interquartile range (i.e., total genome length <4,841,869 bp and >5,523,239 bp. The answer is yes, they did. I took this concern seriously, so I cross-referenced the supplied supplementary data. By doing so, the reviewer will see that these genomes, which would have been considered outliers based on their genome length, belong to clade A (BA1243, BA942, and BA2098) and clade B (BA1287 and BA1408). Therefore, I propose that our results can be trusted as identifying these outliers was not necessary as the downstream analyses focused on clade C strains only – so these clade A and B strains were excluded from our phylogenetic and temporal analyses.

For clarity, I had added the following text to the manuscript:

“However, a few genomes - BA1243, BA942, BA2098 from clade A and BA1287, BA1408 from clade B - exceeded the boundaries of the upper and lower 1.5 x interquartile range for genome length (i.e., total genome length <4,841,869 bp and >5,523,239 bp).”

Minor comments

Line(s) 136-138 While describing the three clades of ST131 it should be noted that clade A is of a different serotype than B and C. This should be added to the text.

Author's Response: We thank the reviewer for this suggestion. The recommendation has been incorporated into the manuscript. I.e., “Clade A (O16:H5 serotype) is the most divergent and primarily defined by the type 1 fimbriae *fimH* adhesin *H41* allele, Clade B (O25b:H4 serotype) is frequently associated with animal to human transmission and primarily defined by *fimH* allele *H22* and Clade C (O25b:H4 serotype) represents the largest ST131 clade and is primarily defined by the *fimH* allele *H30*.”

139-140 Clade C is divided into three sublineages C(0), C1 and C2 e.g. by Ben Zakour *et al* 2016. C0 being susceptible to fluoroquinolones.

Author's Response: We thank the reviewer for this comment. I agree that C0 described in Ben Zakour *et al.* 2016 could be considered a subclade, so I can appreciate and understand the reviewer's comment. But in this instance, I respectfully hold a different perspective. Here, I have written, “Clade C can be further delineated into **two major sub-lineages**: C1 (*H30R*) and C2 (*H30Rx*), **which are resistant to fluoroquinolones**...”. Ben Zakour *et al.* 2016 clearly states “with the three isolates closer to clade B (classified B0) and four isolates closer to clade C (classified C0) illustrating the precise evolutionary events leading to the emergence of clade C”. Ben Zakour *et al.* 2016 later go on to exclude C0 from their SNP analyses (Table 1 & 2) as C0 is described as “intermediate” C0 strains.

143-148 Results from McNally et al 2019 should be added to the list e.g. ST131 has an open pangenome and low genomic fluidity which is rare in *E. coli* and the diversification of certain genetic loci working under the NDFS model.

Author's Response: We thank the reviewer for this suggestion. Added: "...and (v) the open pangenome and low genomic fluidity associated with ST131, which is rare in *E. coli* and the diversification of certain genetic loci working under the negative frequency-dependent selection model."

153-157 Claiming that a guideline recommending the use of betalactams, trimethoprim, cipro or amoxiclav for the treatment of pyelonephritis introduced in 2018 could be contributing to the increasing rates of ESBL-producing *E. coli* in the UK and referring to an article from 2007 (Livermore et al) is not right.

Author's Response: We thank the reviewer for this suggestion. We have corrected the reference to: **Day MJ, Hopkins KL, Wareham DW, Toleman MA, Elviss N, Randall L, et al.** Extended-spectrum β -lactamase-producing *Escherichia coli* in human-derived and foodchain-derived samples from England, Wales, and Scotland: an epidemiological surveillance and typing study. *The Lancet Infectious Diseases*. 2019;19:1325-1335.

180 localised and national transmission? With what reference material this claim can be made?

Author's Response: We thank the reviewer for this comment. We have now removed the wording "with localised or national transmission" from the manuscript.

207-208 (and Table S2) Median doesn't illustrate the situation very well since the extremities are very far from each other. It is good that the range is presented since 0.14 million reads is only 9x coverage. It is good that all isolates under 20x coverage were excluded. It might be worthwhile to present the read counts for the isolates used in the actual analyses.

Author's Response: We thank the reviewer for this comment. In Table S2 and S3, there is a column that denotes the mapped read counts and is named "No. mapped reads (%)". Here, I refer the reviewer to page 1, paragraph 1 of the supplementary appendix, which states: "The average sequence coverage depth was estimated using the Burrows–Wheeler Aligner v0.7.15; SAMtools v1.2; Picard v2.7.1 (<https://github.com/broadinstitute/picard>); the Genome Analysis Tool Kit v3.2-2 (GATK); BEDTools v2.18.2; and SNPEff v4.1 as implemented in SPANDx v3.2. In brief, the trimmed reads were mapped to the complete chromosome of *E. coli* ST131 strain EC958 (GenBank: HG941718)..."

224 Combine Tables S2 and S4. All the stats from the genomes should be in the same table. Now it's difficult to assess the data. There is clear correlation between the N50 and number of contigs as well as N50 and genome size. I think this is an indication of the low coverage and low number of reads, but now it's difficult to compare these numbers to the ones above. When I combined the data myself, it is quite clear that there is a strong correlation with low number of reads and small genome size indicating that the sequencing coverage is too low.

Author's response: We thank the reviewer for raising this concern. Normally, I would agree with this comment; however, to give more context and clarity to this analysis and results, we wish to keep the tables as is. In our workflow, we assessed the read metrics for 157 Welsh isolates (Table S2) before assembly. This identified 15 strains that were removed from further analyses (Table S3). The 142 remaining strains were then *de novo* assembled (Table S4).

228 What does “available datasets were downloaded” mean? There are most likely more than 1000 sequenced ST131 genomes available since McNally et al used almost 900 publicly available genomes in their analyses published in 2019 (BioRxiv 2018).

Author's response: We thank the reviewer for raising this concern. I have amended the text as follows: “available genome sequence data were downloaded including”.

We recognise that including more genomes would have significantly enhanced the robustness of our analysis. However, this study began in 2017, so we were aligning the genomes from Wales with the significant ST131 study conducted by Ben Zakour *et al.* in 2016. Looking back, an alternative option could have been to retrieve genomes from the Enterobase website. However, at the onset of our study, only ~245 *E. coli* ST131 genomes from the United Kingdom were accessible publicly. I've had a look in the dataset and among these, 27 were already included in Ben Zakour *et al.* 2016 and subsequently in our research.

Also, in Table S5 presenting this data, I think it should read *Canis lupus familiaris* not *Canus lumpus familiaris*. Also, some species names e.g., chicken, are not in Latin. Why?

Author's Response: We thank the reviewer for spotting this error in spelling *Canis lupus familiaris*. Regarding the Latin names, I think this is an artefact of using publicly available metadata. Using the reviewer's example, I didn't want to assume that ‘chicken’ was *Gallus gallus*, when it could just as much be *Gallus gallus domesticus*. To correct this, I've changed the heading ‘Source’ to ‘Host’ and replaced the Latin names with the common names.

308-309 Is this only based on the tree likelihoods? The model should be selected based on the stepping-stone or path sampling analyses.

Author's Response: We thank the reviewer for this query. I used a nested sampling Bayesian computation algorithm to test if the strict clock or uncorrelated relaxed clock best fit our dataset, which favoured the uncorrelated relaxed clock model. I also used jmodeltest2 to determine the best-fit models of nucleotide substitution. The Akaike Information Criterion (AIC) and Bayesian Information Criterion (BIC) are popular methods in phylogenetics and make it possible to compare non-nested models. Posada and Buckley (2004) suggest that the lowest AIC or BIC score is selected when testing models and Luo *et al.* have shown that BIC outperforms AIC (albeit for the selection of a substitution model).

Posada D, Buckley TR. Model selection and model averaging in phylogenetics: advantages of Akaike information criterion and Bayesian approaches over likelihood ratio tests. *Syst Biol.* 2004;53:793–808.

Luo A, Qiao H, Zhang Y, Shi W, Ho SYW, Xu W, Zhang A, Zhu C. Performance of criteria for selecting evolutionary models in phylogenetics: a comprehensive study based on simulated datasets. *BMC Evol Biol.* 2010;10:242.

326 Which STs were the two non-ST131 isolates? Please add to the text.

Author's Response: Here, I refer you to Lines 146-148, which state: "All 142 genomes were ST131, except for BA1243 and BA1279 (same Clonal Complex) which differed in the fumarate hydratase class II (*fumC*) and malate dehydrogenase (*mdh*) genes respectively (Figure S3)." Figure S3 denotes key housekeeping genes in red and BLASTn comparisons show discrepancies in these loci. Both of these strains are likely to be ST131, which could be a possible sequencing artefact.

450-451 It's difficult to differentiate the reference from the number. Also, Kallonen et al 2017 should be referred here.

Author's Response: Corrections made and reference added.

527-528 I agree that this study includes highly discriminatory analyses, but no transmission analyses with UK material. Therefore, claiming multiple introductions is too strong a claim.

Author's Response: We thank the reviewer for spotting this.

Text has been amended as follows:

"However, it is crucial to note that this study lacks direct transmission analyses using ST131 genomes from the UK. As a result, while these analyses provide insights into the genomic relationships, the claim of multiple introductions should be considered cautiously without specific transmission analyses involving UK isolates."

562-563 How would inclusion of more isolated from the UK introduce bias to the analyses if you are looking for common ancestors? Temporal maybe, but geographical? In any case, all studies are analysing a small part of the actual bacterial population and including more strains should increase the reliability of the results, not create a bias.

Author's Response: We thank the reviewer for spotting this.

Text has been amended as follows:

"However, while including more ST131 genomes collected from the UK could potentially introduce temporal variations, the aim was to ensure a comprehensive representation for a more accurate understanding of broader evolutionary relationships of ST131. Although more genomes might introduce some temporal variations, geographically diverse samples contribute to a broader representation of the global ST131 population, potentially enhancing the reliability of our findings rather than creating bias. This lack of bias was reflected by our population expansion timeline which coincides with the first reported isolate of ST131 in the UK in 200³⁵, before becoming the predominant clone (n = 52/88, 59.1%) in the Northwest of England (albeit mostly urine-derived strains) between 2004 and 20068. Despite this, expanding the dataset by including additional UK isolates would have indeed increased the breadth of our analysis, aiding in a more comprehensive understanding of within-country transmission dynamics."

565 Even though Kallonen et al. 2017 reported the first ST131 isolate in the UK, that definitely is not the first. They couldn't have been that lucky. Initial detection is therefore too strong a wording. Maybe first reported isolate would be better?

Author's Response: Great spot and I agree. Suggestions incorporated into the manuscript. Thank you.

566 It should be added to the text that Lau et al 2008 were mostly urine isolates.

Author's Response: Added text to reflect this suggestion. "This lack of bias was reflected by our population expansion timeline which coincides with the first reported isolate of ST131 in the UK in 2003, before becoming the predominant clone ($n = 52/88$, 59.1%) in the Northwest of England (albeit mostly urine-derived strains) between 2004 and 2006".

577-578 Claims "Further, this study highlights the need to incorporate whole-genome sequencing with epidemiological data and a 'One Health' approach to identify potential infection reservoirs in the environment". Where does this study highlight the need for a One Health approach? I do not see it anywhere. There are some studies from the UK that would indicate that a One Health approach is not needed e.g. Ludden et al 2019 mBio. I think that this claim should be removed from the text. I agree with incorporating WGS and epidemiological data, but not One Health.

Author's Response: Amended text to reflect this suggestion.

"Further, this study highlights the need to incorporate whole-genome sequencing with epidemiological data to identify potential infection reservoirs in the environment, which will allow for identifying ST131 transmission dynamics between healthcare settings and the community."

Table S2

There is something wrong with the data. For example, on row 146. Under Species #2 is Enterobacteriaceae bacterium0.04. Under No. reads Species #3 (%) it reads Salmonella (285) and under Species #3 it says enterica. Please go through the data to make sure everything is correct.

Author's response: Again, thank you for your thorough review and for spotting my error here. I can confirm that I have gone through the data and that everything is correct.

Reviewer #3 (Remarks to the Author):

White and colleagues report on the genomic epidemiology of UPEC ST131 associated with bacteraemia in Wales between 2013 and 2014. The authors used whole-genome sequencing to characterize circulating ST131 strains and determine their genetic relatedness to previously reported sequence data in Wales and other parts of the world. The objective of the cross-sectional population-based study was clearly expressed and an appropriate methodology, which was explicitly presented to allow for reproducibility, was used. Data analyses and interpretation were properly presented in a consistent manner and the conclusion was well supported by the presented data. The manuscript provides comprehensive valuable data on the evolutionary pattern, geographical clustering and transmission source of circulating ST131 strains which are of public health importance in the prevention of *E.coli* associated bacteraemia as well as antimicrobial resistance.

Authors response: We thank the reviewer for this positive comment about our paper, and we have addressed all comments as suggested.

Major comments

1. The study duration was 12 months which directly impacts the number of *E.coli* blood isolates collected from PHW and Welsh NHS. Did the authors calculate the sample size before collection, and was the study period and sample size adequate for a good representation of the general circulating strains in Wales?

Author's Response: We thank the reviewer for raising this concern. We acknowledge the valid concern raised by the reviewer. We would like to take this opportunity to mention that between April 2013 and March 2014, whole-genome sequencing of *E. coli* from blood isolates was not a standard procedure. This is the first study to include genomes from Wales. So such, this was a restricted dataset from a 12-month survey of bloodstream infections caused by ST131 in Wales. This does not represent *E. coli* associated with UTIs across the Welsh population. Instead, this is an incidence study and not a prevalence study. It is essential to clarify that this manuscript does not represent the general circulating strains in Wales. Instead, it describes some of the first sequenced *E. coli* ST131 genomes associated with bloodstream infections, with subsequent genomic analyses. Future studies would benefit from sample size calculations to get a better representation of the general circulating strains in Wales.

2. The study limitations were not highlighted in the manuscript.

Authors response: We thank the reviewer for this comment about our paper. We have now incorporated the following into the manuscript:

“The absence of ST131 genomes from the UK in this study is acknowledged as a limitation. Although including more ST131 genomes from the UK could have strengthened our analysis, the study was started in 2017 and relied on available data, notably from a significant ST131 study in 2016¹⁰. Despite limited publicly accessible UK ST131 genomes (~245), a subset was already incorporated in our research from a follow-up study⁴⁰ to Ben Zakour et al.¹⁰. We acknowledge this limitation's impact on the study's robustness, and exploring additional data sources initially could have bolstered our analysis of within-country transmission dynamics.

Minor comment

Line 334, the frequencies of clade A (15.5%) and clade B (12.0%) are not similar. It would be more appropriate to state there was no significant difference between both.

Author's Response: Thank you for this insightful comment. According to your comment, we have revised the wording to:

“While most isolates were located within clade C ($n = 103/142$, 72.5%), there was no substantial difference between representatives from clade A ($n = 22/142$, 15.5%) and clade B ($n = 17/142$, 12.0%).”

REVIEWERS' COMMENTS

Reviewer #2 (Remarks to the Author):

My concerns have been adequately addressed.